

# Knowledge, attitude, and practices of front line health workers after receiving a COVID-19 vaccine: a cross-sectional study in Pakistan

Sadia Minhas[1], Aneequa Sajjad[1], Iram Manzoor[2], Atika Masood[3], Agha Suhail[4], Gul Muhammad Shaikh[5] and Muhammad Kashif[6]

[1] Oral Pathology, Akhtar Saeed Medical and Dental College, Lahore, Punjab, Pakistan
[2] Community Medicine, Akhtar Saeed Medical and Dental College, Lahore, Punjab, Pakistan
[3] Pathology, Akhtar Saeed Medical and Dental College, Lahore, Punjab, Pakistan
[4] Operative Dentistry, Akhtar Saeed Medical and Dental College, Lahore, Punjab, Pakistan
[5] Dental Education and Research, Shahida Islam Medical and Dental College, Lodhran, Punjab, Pakistan
[6] Oral Pathology, Bakhtawar Amin Medical and Dental College, Multan, Punjab, Pakistan

Corresponding author
Sadia Minhas,
sadiawasif81@gmail.com

## ABSTRACT

**Background:** Globally, there is an increased risk of COVID-19 infection among front-line health workers (FHW). This study aimed to evaluate the knowledge, attitude, and practices of FHW of Pakistan after receiving the COVID-19 vaccine.
**Methods:** A population web-based survey on COVID-19 vaccine was conducted on 635 FHW in Pakistan between April 15, 2021, and July 15, 2021. The survey focused on four main sections consisting of socio-demographic data, knowledge, attitude, and practices after receiving the COVID-19 vaccine. The data was analyzed on SPSS. $p < 0.05$ was considered significant.
**Results:** Overall, 60% of FHW were nervous before getting vaccinated, with the leading reason to get vaccinated being their concern to protect themselves and their community (53.4%). A majority of FHW had fear about the unseen side effects of the COVID-19 vaccine (59.7%) used in Pakistan, with the most common side effect reported as soreness at the injection site (39%). It has been noted that almost all of the FHW observed preventive practices after getting vaccinated. The results showed that married respondents had favorable practices towards COVID-19 vaccines (B = 0.53, $p < 0.01$) (B, unstandardized regression coefficient). It was also found that more informational sources (B = 0.19, $p < 0.01$), higher knowledge of vaccination (B = 0.15, $p < 0.001$), and favorable attitude toward vaccine (B = 0.12, $p < 0.001$) significantly predicted favorable practices toward COVID-19 vaccination.
**Conclusion:** The findings reflect that FHW, though they were worried about its side effects, have good knowledge and a positive attitude after getting the COVID-19 vaccine. This study is significant as the FHWs are a symbol for guidance, a reliable source of information, and an encouraging means of receiving COVID-19 vaccine for the general public. This study also reported that post-vaccination side effects were mild which will aid in reducing the vaccine hesitancy among the general Pakistani population.

## INTRODUCTION

Globally, all parts of life have been affected because of the COVID-19 pandemic (*Huynh et al., 2021*). According to the World Health Organization (WHO), there have been 308,458,509 confirmed cases of COVID-19, including 5,492,595 deaths, and a total of 9,194,549,698 vaccine doses have been administered worldwide until January 11, 2022 (*WHO, 2022a*). WHO has listed FHWs as a priority group for COVID-19 vaccination (*Gagneux-Brunon et al., 2021*), as they are at increased risk because of their direct contact with patients with COVID-19 (*Dhahri, Iqbal & Khan, 2020*). Initially, 10–20% of FHW have been identified with the COVID-19 infection; therefore, protecting them from COVID-19 infection plays a vital role in the conservation of the healthcare system (*Nguyen et al., 2020*).

The COVID-19 infection is extremely contagious and involves the worldwide population; therefore, the most useful approach to protect the population from COVID-19 infection is vaccination, which is a significant public health measure (*Cascella et al., 2022*). Ten COVID-19 vaccines have been registered by WHO and distributed in different countries so far, with almost 33 vaccines being approved by at least one country worldwide (*COVID-19 Vaccines, 2022*). The FHWs are suggested as the top priority for vaccination in contrast to the general population (*Malik, Malik & Ishaq, 2021*; *WHO, 2022b*). Therefore, it is significant for them to attain early vaccination coverage that will assure a sufficient workforce to treat the infected patients (*Thorsteinsdottir & Madsen, 2021*).

The accessibility of COVID-19 vaccines might be the only means to control the COVID-19 infection, as, in several countries, long-term lockdown is not possible because of economic crisis. Thus, in addition to following the COVID-19 Standard Operating Procedures (SOP), there is an essential need to be vaccinated against COVID-19 infection to limit the COVID-19 community transmission in Pakistan (*Malik, Malik & Ishaq, 2021*). To evaluate the beliefs and intentions about the past vaccination, the health belief model is a beneficial method that evaluates perceived susceptibility and severity, perceived barriers, benefits, and signs of action (*Giao et al., 2019*). Regarding the safety of the COVID-19 vaccine, the doubts of FHW must be satisfied soon, as they are the first ones to receive the vaccine (*Huynh et al., 2021*). Literature search has revealed that percentages of knowledge and acceptance of COVID-19 vaccines in FHW vary among countries, which were 73.9% in Europe, 40% in Hong Kong, and only 27.7% in the Congo, accordingly (*Nzaji et al., 2020*; *Wang et al., 2020b*). Globally, the prevalence of COVID-19 vaccination hesitancy among FHW ranged from 4.3% to 72% which varies because of several socio-demographic features (*Biswas et al., 2021*). To develop trust among FHW, self-governing committees and trusted bodies should deliver reliable knowledge and information regarding COVID-19 vaccines and their safety (*Kasozi et al., 2021*). Currently, many vaccines are available that are believed to be safe and effective, though doubts in evaluating the efficacy of these vaccines still exist (*Ledda et al., 2021*). In Pakistan, WHO has declared that there

have been 1,345,801 confirmed cases of COVID-19 with 29,042 deaths, and that 169,131,246 vaccine doses have been administered until January 21, 2022 (*WHO, 2022b*). Currently, several rumors regarding the COVID-19 vaccine are spreading in Pakistan, with these rumors including that, with the vaccine, nano-chips are inserted in the bodies to achieve control of the individuals through 5G towers (*Khan et al., 2020*), in addition, that the vaccine was a magnificent trick to target the Islamic nations and was formed to allow the Jews to take control of the world.

Accordingly, FHW play a significant role both in delivering knowledge regarding the source of COVID-19 vaccine and its effects in the upcoming years, as well as in serving as role models for the general population in encouraging them to get vaccinated against COVID-19 infection. As data on the knowledge, attitude, and practices after getting vaccinated against COVID-19 among FHW of Pakistan are rare, this study aims to assess the role of socio-demographic characteristics, knowledge, and attitude of FHW in predicting practices after getting the COVID-19 vaccine. In addition, it aims to develop policies with the help of this survey to have a helpful and continuous vaccination rollout plan for COVID-19 infection by the government of Pakistan.

## MATERIALS AND METHODS

### Design

A cross-sectional study was conducted to assess the knowledge, attitude, and practices of FHW of Pakistan after getting the COVID-19 vaccine from April 15, 2021, to July 15, 2021. Because of the third wave of COVID-19 in Pakistan, it was not possible to perform the community-based survey; therefore, a semi-standardized electronic questionnaire was designed to collect the data using an online secure Google survey tool, and a shareable link was produced (https://docs.google.com/forms/d/1YCUf42WO3X2qS6hf36tw_6x6Ni5eAS4YZVn6W_L33ks/edit) and disseminated through multiple social media such as a personal social network, Facebook, and WhatsApp with mandatory email addresses to ensure one response per individual so that results may not be compromised. The survey includes an introduction that specified the purpose of the study. The data were collected anonymously, and email addresses were kept confidential. The biasness of the study with respect to participation was decreased by keeping the survey open for 3 months which provided ample time for people to participate in the survey.

### Ethical declaration

The current study was conducted following the principles for human investigations (*i.e.*, Helsinki Declaration) and has passed the ethical approval from the institutional review board of Akhtar Saeed Medical and Dental College (M-21/069/-Oral Pathology). All the respondents participated willingly in the survey. The respondents were assured that their information will be kept confidential.

### Inclusion and exclusion criteria

The inclusion criteria of the study participants were being a Pakistani resident, FHW, and having internet access. The exclusion criteria included an incomplete survey.

## Sample size

The sample size was calculated using the following equation:

$$n = \frac{z^2 pq}{d^2}$$

$$n = \frac{1.99^2 \times 5 \times (1 - 0.5)}{0.05^2}$$

$$n = 396.01$$

where,

$n$ = number of samples

$z$ = 1.96 (95% confidence level)

$p$ = prevalence estimate (50% or 0.5%) (*Islam et al., 2021*)

$q$ = (1 2 $p$)

$d$ = precision limit or proportion of sampling error (0.05).

As there is no earlier similar study on FHW concentrating on knowledge, attitudes, and practices after getting the COVID-19 vaccine in Pakistan, the best assumption ($p$) made for the present study would be 50%. A sample size of 396.01 participants was assessed, assuming a 10% non-response rate. Our sample size exceeded this estimate.

## Questionnaire

A 30-item online self-administered questionnaire was made from earlier studies according to the objectives of the current study (*Danabal et al., 2021*; *Islam et al., 2021*). The Google form consists of four key focus areas, including socio-demographic data, knowledge about COVID vaccine, post-vaccination attitude, and practices along with informed consent. Each section consists of a range of options from multiple choices to forced-choice (yes/no) questions depending on the subject matter. All the questions were compulsory. In the first section, respondents were asked nine questions regarding their socio-demographic characteristics such as age and gender. The key independent variables were knowledge and attitude related to the COVID-19 vaccine. Knowledge of the COVID-19 vaccine was assessed using four items with three possible responses (*i.e.*, Yes, No, and Don't know). A sample item used to assess knowledge included: Vaccine is important to end the COVID-19 pandemic. A higher score on the scale showed higher knowledge of the COVID-19 vaccine. Attitude toward the COVID-19 vaccine was assessed using 10 items with five possible responses (*i.e.*, *Strongly disagree*, *Disagree*, *Neutral*, *Agree*, and *Strongly agree*), which were later recoded to three responses (*Disagree*, *Neutral*, and *Agree*). A sample item used to assess attitudes included: Do you believe that the benefits of COVID-19 vaccination are greater in comparison to its risk? A higher score on the scale showed favorable attitudes toward COVID-19 vaccine. The dependent variable included in the study was practiced after getting the COVID-19 vaccine, which was assessed using six items with three possible responses (*i.e.*, Yes, No, and Don't know). A sample item used to assess practices included: Do you still follow COVID-19 basic prevention guidelines after

getting vaccinated? A higher score on the scale showed favorable practices toward COVID-19 vaccine.

## Participants

The target population of the current study was FHW, including physicians, dental surgeons, pharmacists, physiotherapists, laboratory technicians, nurses, hospital administrative staff, and undergraduate medical and allied health sciences students from all over Pakistan. A total of 635 responses were obtained.

## Statistical analysis

The data cleaning, editing, and sorting were done on the Microsoft Excel version (2014). The data were then imported on SPSS version 20 (IBM Corp., Armonk, NY, USA) spreadsheet where the coding of data and analysis was done. The quantitative data are presented in the form of descriptive statistics as frequencies and percentages. First-order analyses such as the chi-square test and Fisher exact test were completed to check the association among socio-demographic characteristics (age and gender), attitude, and practices after COVID-19 vaccine.

To predict practices after getting the COVID-19 vaccine from socio-demographic and key independent variables (knowledge and attitude), a multivariate linear regression analysis was conducted. To test for multicollinearity, tolerance values and VIF ( ) values were considered which showed that multicollinearity was not a threat. A $p$-value <0.05 was considered statistically significant with a 95% confidence interval.

# RESULTS

## Demographic data

A total of 704 respondents from all over Pakistan participated in this study, out of which 69 were excluded because of an incomplete responses, and thus, the response rate for the study was 90.2%. Table 1 gives an outline of their socio-demographic characteristics. Most of the respondents ($N = 296$, 46.3%) were young (20–30 years), females (373, 58.7%), single (349, 55%), and completed a bachelor degree in dental surgery (289, 45.5%). The majority of the participants collected information about the COVID-19 vaccine *via* social media and television (241, 38%). The most frequently reported minor side effects were soreness at the injection site ($N = 246$, 39%), followed by body aches ($N = 95$, 15%) (Fig. 1).

## Information regarding COVID-19

The results showed that, with more than half of the respondents (60%) stating that they were nervous before getting the COVID-19 vaccine, 31% of the respondents had been diagnosed previously with COVID-19 infection.

## Data regarding COVID-19 vaccine knowledge

The participant's responses regarding knowledge about the COVID-19 vaccine are summarized in Table 2. The most common reason for getting the COVID-19 vaccination was their concern to protect themselves and their community (53.4%). Intriguingly, 88.5% were aware that people can catch COVID-19 infection even if they are vaccinated.

**Table 1 Descriptive statistics of socio-demographic characteristics.**

| Variables | N (%) |
|---|---|
| Age | |
| 20–30 | 294 (46.3) |
| 31–40 | 207 (32.6) |
| 41–50 | 80 (12.6) |
| 51–60 | 25 (3.9) |
| Above 60 | 29 (4.6) |
| Gender | |
| Male | 262 (41.3) |
| Female | 373 (58.7) |
| Marital status | |
| Married | 286 (45) |
| Single | 349 (55) |
| Education level | |
| MBBS | 223 (35.1) |
| BDS | 289 (45.5) |
| Pharmacy | 39 (6.1) |
| Physiotherapy (DPT) | 17 (2.7) |
| Graduation | 51 (8) |
| Under graduation | 16 (2.5) |
| Occupation | |
| Physician | 136 (21.4) |
| Dental surgeon | 182 (28.7) |
| Pharmacist | 37 (5.8) |
| Physiotherapist | 17 (2.7) |
| Laboratory technician | 30 (4.7) |
| Nurse/Dispenser | 15 (2.4) |
| Hospital administrative | 30 (4.7) |
| Medical student | 188 (29.6) |
| Are you currently in practice? | |
| Yes | 314 (49.4) |
| No | 321 (50.6) |
| Where did you receive information about the COVID-19 vaccine? | |
| Social media (WhatsApp, Facebook) | 120 (18.9) |
| Friends and family | 56 (8.8) |
| Television | 26 (4.1) |
| Workplace | 31 (4.9) |
| Government helplines | 22 (3.5) |
| SM, NP, WP | 24 (3.8) |
| Friends, government helplines | 13 (2.0) |
| Social media and television | 241 (38.0) |
| Social media, friends and family | 37 (5.8) |

| Table 1 (continued) | |
| --- | --- |
| **Variables** | **N (%)** |
| All | 65 (10.2) |
| When did you get vaccinated against COVID-19? | |
| January 2021 | 19 (3) |
| February 2021 | 77 (12.1) |
| March 2021 | 150 (23.6) |
| April 2021 | 206 (32.4) |
| May 2021 | 69 (10.9) |
| June 2021 | 114 (18) |

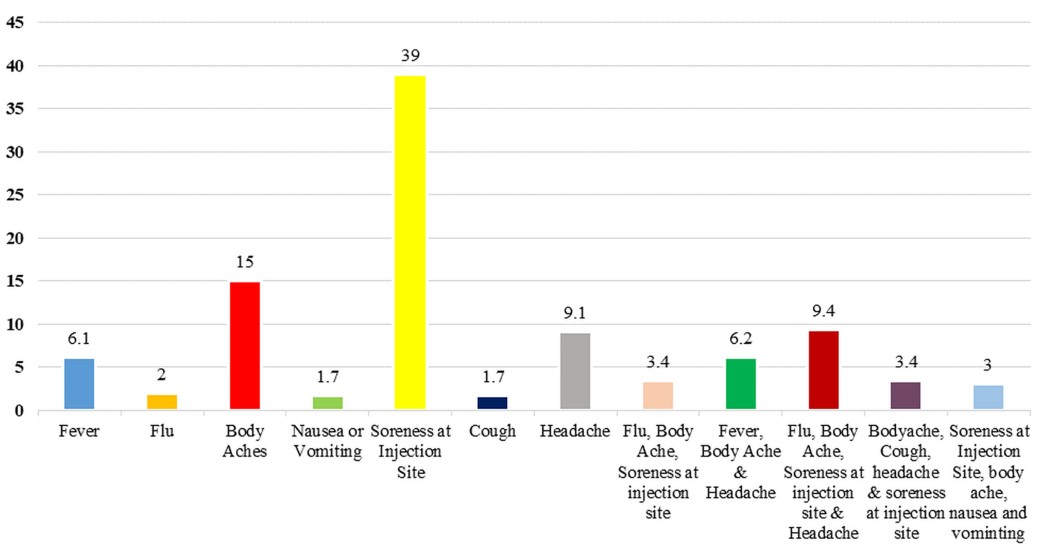

**Figure 1** Minor side effects of respondents after receiving COVID-19 vaccine.

## Data regarding belief and attitude about COVID-19 vaccine

Table 3 shows the distribution of participants regarding the belief and attitudes of FHW after getting the COVID-19 vaccine. Of 635 respondents, 356 (56%) agreed that the newly developed COVID-19 vaccine is not safe as it is developed in an emergency. However, despite their concerns regarding the unseen effects of the COVID-19 vaccines (59.7%), most of the respondents still believed that it will not discourage them from getting vaccinated (84.5%). When it comes to trust in the measures taken by the Ministry of Health of Pakistan to ensure vaccine safety, only 59 (9.3%) did not have trust. The responses of participants are summarized in Table 3.

## Practices after getting COVID-19 vaccine

The frequencies of practices among the respondents after getting the COVID-19 vaccine are presented in Table 4. It was observed that a large number of participants (95%) showed

**Table 2 Frequencies and percentages regarding knowledge about COVID-19 vaccine.**

| Variables | N (%) |
|---|---|
| Which of the following best describes your reason for getting vaccinated against COVID-19? | |
| I want to protect myself and the community | 339 (53.4) |
| I have increased exposure to the patients | 65 (10.2) |
| I had COVID in the past and don't want to have it again | 47 (7.4) |
| It can lower the chance of getting COVID-19 | 74 (11.7) |
| To be able to travel safely | 19 (3.0) |
| It helps to stop the COVID-19 pandemic | 51 (8.0) |
| It is a safer way to help build the protection | 40 (6.3) |
| How long will it take to build immunity against COVID-19 after getting vaccinated? | |
| Immediately after the first dose | 16 (2.5) |
| 14 days after the first shot | 141 (22.2) |
| Immediately after the second dose | 47 (7.4) |
| 14 days after the second dose | 310 (48.8) |
| One to two months after the second dose | 121 (19.1) |
| Have you ever been infected with COVID-19 infection? | |
| Yes | 197 (31.0) |
| No | 321 (50.6) |
| Don't know | 117 (18.4) |
| Do you feel nervous before getting COVID-19 vaccination? | |
| No | 254 (40) |
| Yes | 381 (60) |
| After getting a COVID-19 vaccine can you still catch COVID-19 infection and can also infect others | |
| Yes | 482 (75.9) |
| No | 55 (8.6) |
| Don't know | 98 (15.4) |
| A vaccine is important to end the COVID-19 pandemic | |
| Yes | 562 (88.5) |
| No | 38 (6) |
| Don't know | 35 (5.5) |

a positive attitude toward the preventive COVID-19 practices. The COVID-19 vaccine acceptance is related to the idea of recommending the vaccine to friends and family. Of note, a large majority of the respondents (93.9%) will recommend the COVID-19 vaccine to their friends and family. Females ($N = 181$, 58.4%) were more knowledgeable regarding the duration to build the immunity than males ($N = 129$, 41.6%), with a significant association between them ($p = 0.034$). There were significant differences in belief and attitude after getting the COVID-19 vaccine among both genders. It was observed that females ($N = 112$, 72.3%) were significantly more nervous before getting vaccinated than males ($N = 43$, 27.7%). Furthermore, female respondents were more likely to follow COVID-19 practices than males. It was observed that 50% of females wear a mask

**Table 3 Belief and attitude of respondents after getting COVID-19 vaccine (N = 635).**

| Variables | Agree N (%) | Disagree N (%) | Neutral N (%) |
|---|---|---|---|
| Safety of a vaccine cannot be considered guaranteed | 356 (56) | 75 (11.8) | 204 (32.2) |
| Worried about unseen effects of COVID-19 vaccine | 379 (59.7) | 86 (13.5) | 170 (26.8) |
| Believe that the side effects of the vaccine are reasonable and will not discourage me from taking the vaccine | 537 (84.5) | 31 (4.9) | 67 (10.6) |
| After getting a COVID-19 vaccine you are worried about getting the virus | 360 (56.7) | 149 (23.5) | 126 (19.8) |
| Trust the Ministry of Health of Pakistan regrading vaccine safety | 490 (77.2) | 59 (9.3) | 86 (13.5) |
| Concerned about COVID-19 vaccine efficacy | 434 (68.4) | 84 (13.2) | 117 (18.4) |
| Believe that benefits of COVID-19 vaccine are greater than its risk | 542 (85.4) | 29 (4.5) | 64 (10.1) |
| Believe that COVID-19 vaccine provides long term protection | 269 (42.3) | 177 (27.9) | 189 (29.8) |
| Believe that COVID-19 vaccine is approved quickly | 311 (48.9) | 184 (28.9) | 140 (22.2) |
| Have you ever had serious reaction after getting COVID-19 vaccine | 11 (1.8) | 624 (98.2) | 0% |

**Table 4 Detailed distribution of practices of respondents.**

**Follow basic SOPS after getting COVID-19 vaccine**

| Responses | Respondents N (%) |
|---|---|
| Yes | 603 (95) |
| No | 17 (2.7) |
| Don't know | 15 (2.3) |
| Recommend COVID-19 vaccine to relatives and friends | |
| Yes | 596 (93.8) |
| No | 15 (2.4) |
| Don't know | 24 (3.8) |
| Recommend COVID-19 vaccine to immuno-compromised patients | |
| Yes | 367 (57.8) |
| No | 141 (22.2) |
| Don't know | 127 (20) |
| Still wear a face mask after COVID-19 vaccination | |
| Yes | 614 (96.7) |
| No | 8 (1.3) |
| Don't know | 13 (2) |
| Have you completed your dosage of COVID-19 vaccine? | |
| Yes | 343 (54) |
| No | 292 (46) |

in comparison to males (40.4%) after getting the COVID-19 vaccine, which was found to be significantly associated ($p = 0.046$) (Table 5).

Out of 197 respondents who had a previous history of COVID-19 infection, it was observed that most of them were between 20 and 30 years old ($N = 83$, 42.1%), followed by

**Table 5 Effects of genders on respondents belief and attitudes after getting COVID-19 vaccine by applying Chi-square and Fisher exact test.**

| Gender | Strongly agree | Agree | Strongly disagree | Disagree | Neutral | $p > 0.05$ |
|---|---|---|---|---|---|---|
| Worry about unseen side effects of COVID-19 vaccine | | | | | | |
| Males | 38 (14.5) | 116 (44.3) | 8 (3.1) | 30 (11.5) | 70 (26.7) | 0.901 |
| Females | 53 (14.2) | 172 (46.1) | 7 (1.9) | 41 (11) | 100 (26.8) | |
| Concerned about vaccine efficacy | | | | | | |
| Males | 63 (24) | 109 (41.6) | 8 (3.1) | 39(14.9) | 43 (16.4) | 0.040 |
| Females | 88 (23.6) | 174 (46.6) | 4 (1.1) | 33 (8.8) | 74 (19.8) | |
| Believe that vaccine side-effects are reasonable | | | | | | |
| Males | 83 (31.7) | 136 (51.9) | 0 | 11 (4.2) | 32 (12.2) | 0.037 |
| Females | 90 (24.1) | 228 (61.1) | 5 (1.3) | 15 (4) | 35 (9.4) | |
| Trust the ministry of health of Pakistan regarding vaccine safety | | | | | | |
| Males | 68 (26) | 151 (57.6) | 5 (1.9) | 12 (4.6) | 26 (9.9) | 0.000 |
| Females | 52 (13.9) | 219 (58.7) | 9 (2.4) | 33 (8.8) | 60 (16.1) | |
| Benefits are greater than risk | | | | | | |
| Males | 85 (32.4) | 144 (55) | 4 (1.5) | 9 (3.4) | 20 (7.6) | 0.064 |
| Females | 87 (23.3) | 226 (60.6) | 7 (1.9) | 9 (2.4) | 44 (11.8) | |
| COVID-19 vaccine provide long term protection | | | | | | |
| Males | 27 (10.3) | 96 (36.6) | 10 (3.8) | 66 (25.2) | 63 (24) | 0.002 |
| Females | 16 (4.3) | 130 (34.9) | 5 (1.3) | 96 (25.7) | 126 (33.8) | |

31–40 years old ($N = 71$, 36%), with a positive correlation among them ($p = 0.036$). When participants have inquired about the reason for getting vaccinated, it was seen that a majority of the respondents of 20–30 years old ($N = 161$, 47.5%) want to protect themselves and their community ($p = 0.017$). The level of nervousness before getting vaccinated was significantly associated ($p = 0.000$) among different age groups, *i.e.*, decreasing with increasing age predominantly seen in 20–30 years ($N = 71$, 45.8%), followed by 31–40 years ($N = 61$, 39.4%), 41–50 years ($N = 18$, 11.6%), 51–60 years ($N = 2$, 1.3%), and above 60 years ($N = 3$, 1.9%) accordingly. Regarding practices among different age groups, it was found that all age groups followed the basic SOP of COVID-19 infection and individuals above 41 years old (100%) were more likely to wear a mask than those of younger age groups after their vaccination. The model predicting favorable attitudes after getting the COVID-19 vaccine was statistically significant, $F (13, 621) = 4.609$, $p < 0.001$, adj. $R^2 = 0.069$. The value of adjusted $R^2$ showed that approximately 7% of the variation in favorable attitudes after getting the COVID-19 vaccine was predicted by the independent variables of the study. The results showed that favorable attitudes after getting the COVID-19 vaccine increased with age by 0.37 but the change was statistically insignificant. Another finding of the study was that the favorable attitudes after getting the COVID-19 vaccine increased by 0.46 units in those respondents who were not currently practicing but it was statistically insignificant. The results further showed that with a one unit increase in knowledge of respondents for the COVID-19 vaccine, favorable attitudes after getting the

**Table 6 Multivariate regression analyses for models predicting attitudes and practices towards COVID-19 vaccine ($N = 635$).**

| Variables | Attitudes towards COVID-19 vaccine[†] | | | | | Practices towards COVID-19 vaccine[‡] | | | | |
|---|---|---|---|---|---|---|---|---|---|---|
| | B | SE | t | β | p-value | B | SE | t | β | p-value |
| Age[a] | 0.29 | 0.11 | 2.67 | 0.12 | **0.008** | 0.03 | 0.07 | 0.44 | 0.02 | 0.660 |
| Gender[b] | −0.31 | 0.21 | −1.48 | −0.06 | 0.141 | −0.22 | 0.13 | −1.69 | −0.07 | 0.092 |
| Marital status[c] | 0.01 | 0.24 | 0.02 | 0.00 | 0.983 | 0.53 | 0.15 | 3.43 | 0.16 | **0.001** |
| Education[d] | −0.26 | 0.33 | −0.79 | −0.03 | 0.428 | −0.02 | 0.21 | −0.11 | −0.01 | 0.912 |
| Occupation[e] | 0.23 | 0.30 | 0.76 | 0.04 | 0.445 | −0.05 | 0.19 | −0.27 | −0.01 | 0.789 |
| Currently practicing[f] | 0.55 | 0.25 | 2.21 | 0.11 | **0.028** | −0.05 | 0.16 | −0.33 | −0.02 | 0.742 |
| Month of vaccination[g] | −0.06 | 0.08 | −0.80 | −0.03 | 0.426 | −0.03 | 0.05 | −0.68 | −0.03 | 0.497 |
| Information sources | 0.07 | 0.10 | 0.67 | 0.03 | 0.505 | 0.19 | 0.07 | 2.86 | 0.11 | **0.004** |
| Knowledge | 0.24 | 0.06 | 4.45 | 0.17 | **<0.001** | 0.15 | 0.04 | 4.23 | 0.16 | **<0.001** |
| Attitude | — | — | — | — | — | 0.12 | 0.03 | 4.61 | 0.18 | **<0.001** |

Notes:
B, unstandardized regression coefficient; SE, Standard error; β, Standardized regression coefficient.
[a] $1 = 20–30$, $2 = 31–40$, $3 = 41–50$, $4 = 51–60$, $5 =$ Above 60.
[b] $1 =$ Male, $2 =$ Female.
[c] $1 =$ Single, $2 =$ Married.
[d] $1 =$ Graduation/Undergraduate, $2 =$ Minimum 16 years with medical specialization.
[e] $1 =$ Medical student, $2 =$ Medical professional.
[f] $1 =$ Yes, $2 =$ No.
[g] $1–6 =$ January–June 2021. Bold and italic indicates significant.
[†] $F_{(9,625)} = 4.599$; $p < 0.001$; $R^2_{Adj.} = 0.049$.
[‡] $F_{(10,624)} = 9.507$; $p < 0.001$; $R^2_{Adj.} = 0.118$.

COVID-19 vaccine increased by 0.34 units (B = 0.34, $p < 0.001$). Favorable attitude toward COVID-19 vaccine increased by 0.85 units in respondents who were nervous before getting COVID vaccine (B = 0.85, $p < 0.01$). The remaining variables included in the model were statistically insignificant ($p > 0.05$) (Table 6). Table 6 shows that the model predicting favorable practices after getting the COVID-19 vaccine was statistically significant, $F$ (14, 620) = 10.10, $p < 0.001$, adj. $R^2$ = 0.167. The value of adjusted $R^2$ showed that 16.7% of the variation in favorable practices toward COVID-19 vaccine was predicted by the independent variables of the study. The results of multivariate regression analysis further showed that females were less likely to have favorable practices after getting the COVID-19 vaccine than males though the relationship was statistically insignificant (B = −0.24, $p > 0.05$). A significant finding of the study was that the favorable practices toward COVID-19 vaccine increased by 0.49 units in married respondents (B = 0.49, $p < 0.01$). Another significant finding of the study was that the favorable practices toward vaccines increased by 0.16 units with one unit increase in sources of information (B = 0.16, $p < 0.05$). This shows that respondents with multiple sources of information adopted favorable practices after getting the COVID-19 vaccine compared to those with fewer sources of information. The results also showed that those who had been previously infected with COVID-19 displayed poor practices toward the COVID-19 vaccine (B = −0.55, $p < 0.01$). Both key independent variables included in the study were statistically significant. The results showed that with a one unit increase in knowledge of respondents with respect to COVID-19 vaccine, favorable practices after getting the COVID-19 vaccine increased by 0.34 units (B = 0.34, $p < 0.001$). Furthermore, one unit increase in favorable attitudes after

getting the COVID-19 vaccine increased favorable practices toward COVID-19 vaccine by 0.10 units (B = 0.10, $p < 0.001$).

## DISCUSSION

COVID-19 vaccine is an ideal way to end the COVID-19 infection. The government of Pakistan has already started the COVID-19 vaccination rollout in February 2021 creating hope as part of the pandemic solution (*Siddiqui et al., 2021*). To date, very limited studies have investigated the response of FHWs regarding their knowledge and attitude after getting vaccinated. Playing an important role in the general population vaccine acceptance, this study provides first-hand and novel information about the knowledge, attitude, and practices of FHW in Pakistan after getting the COVID-19 vaccination. Important in developing health-related education and awareness programs, the results show that a few socio-demographic, knowledge, and attitude variables influence the practices after getting the COVID-19 vaccine. Currently, social media plays an important role in creating awareness and spread of messages across the masses. In our study, the majority of the participants have acquired knowledge from social media and television regarding the importance of the COVID-19 vaccine (*Alwi et al., 2021*; *Shekhar et al., 2021*). In addition, the government of Pakistan has taken all the appropriate steps for providing facts and figures regarding COVID-19 prevention and recommendations to the public, where a study from Saudi Arabia stated that the majority of the participants obtain information about the COVID-19 vaccine through social media (*Othman et al., 2022*). Mobile applications related to healthcare and informative television campaigns have played an immense role in creating awareness about vaccination programs and are also useful for policymakers regarding the COVID-19 vaccine information in Pakistan.

There is a lot of debate on social media regarding vaccine safety, its adverse effects, effectiveness, and approval by authorities. It has created a number of concerns and a large amount of anxiety even in FHW. More than 60% of the respondents were nervous before getting the vaccine, which is consistent with previous studies (*Malik et al., 2020*; *Shekhar et al., 2021*). This might be because of a high level of concern as there is clear uncertainty regarding COVID-19 safety. The new technique involves mRNA-based vaccines, which have been developed in an emergency, *i.e.*, less than a year has created doubts among the general population, thereby decreasing the acceptance rate (*El-Elimat et al., 2021*). Despite these concerns, 84.5% of FHWs were willing to get the vaccine, which might be related to good knowledge about the severity of COVID-19 infection and trust in the health ministry of Pakistan, which is in accordance with the studies conducted in Pakistan, Saudi Arabia, and France (*Detoc et al., 2020*; *Tahir et al., 2021*), who found that the majority of FHW expressed willingness for the COVID-19 vaccine (*Barry et al., 2020*). Likewise, the present study is in line with the study conducted in Australia (*Dodd et al., 2021*), where concern about the vaccine safety was 36%, and 85.5% were willing to get the vaccination. However, vaccine hesitancy is fueled by anti vaccinationists because of the new technology and the short duration of vaccine development. Such misleading information on social media can shape participants' refusal or acceptance of the COVID-19 vaccine.

In our study, 59.7% of the participants were worried that the vaccine currently being used in Pakistan could have some side effects, which are similar to the national and international data (*Callaghan et al., 2020.*; *Elharake et al., 2021*; *Malik, Malik & Ishaq, 2021*). Literature search has revealed that the leading cause for vaccine hesitancy was fear of the adverse effects of the COVID-19 vaccines (*Callaghan et al., 2020*; *Wang et al., 2020a*; *Alwi et al., 2021*; *Elharake et al., 2021*; *Malik, Malik & Ishaq, 2021*). Nearly all reported side effects were mild, and with there being multiple conspiracy theories linked with the vaccine efficacy and its associated side effects, most frequently reported minor side effect is shown to be soreness at the injection site, which is in agreement with another local study (*Siddiqui et al., 2021*). Therefore, it is significant to take relative measures to reduce such rumors that will ultimately increase the willingness to get vaccinated (*Shekhar et al., 2021*; *Pogue et al., 2020*).

Misconception and lack of trust are contributing factors to COVID-19 vaccine acceptance. The finding of this study (77.2%) is aligned with the study conducted by *Benis, Seidmann & Ashkenazi (2021)* reflecting that the majority of participants have complete trust in their government regarding the COVID-19 vaccine. Data from previously performed local and international studies revealed that a population with an increased level of trust is linked with a high rate of vaccine acceptance (*Vasilevska, Ku & Fisman, 2014*; *Qamar et al., 2021*). Such a high level of trust in the government reflects good policies of the government in handling the pandemic.

Age, currently in practice, and knowledge were associated factors that determined the attitude of FHW after getting the COVID-19 vaccine (Table 6). The current study results showed that the older age group FHW had more positive attitudes than the younger age groups, which is in contrast with the studies from Pakistan and Uganda (*Olum et al., 2020*; *Rehman et al., 2021*). FHW who are currently in practice were more likely to have a favorable attitude even after getting the COVID-19 vaccine as these FHW were more prone to the infection than those who were not in practice, where this discrepancy on attitudes may be because older individuals have an increased risk of exposure to COVID-19 infection in contrast to young individuals, which is in line with the study conducted in Ethiopia (*Ahmed et al., 2021*). Therefore, it can be concluded that FHWs who were old and practicing had a more favorable attitude after getting the COVID-19 vaccine.

In the present study, females showed a less likely positive attitude after getting the vaccine and prevention, which is in contrast with the studies conducted in Bangladesh (*Ferdous et al., 2020*) and Indonesia (*Harapan et al., 2016*) but in accordance with the study conducted in China (*Wang et al., 2020a*). Less attitude and prevention after getting the COVID-19 vaccine poses females at increased risk of acquiring the infection; therefore, importance should be given to maintaining precautions for this specific gender to control the spread of COVID-19 more effectively.

Overall, good hygiene practices and a positive attitude toward preventive measures were observed among the participants of the current study, which is in accordance with the earlier reported data on FHW from Pakistan and Vietnam (*Huynh et al., 2020*; *Saqlain et al., 2020*; *Ladiwala et al., 2021*) but in contrast to the study conducted in Uganda (*Kamacooko et al., 2021*). With the current study showing that knowledge and awareness

play a significant role in the practices to prevent the risks of infection with COVID-19, good hygiene practices and preventive measures in FHW can be attributed to proper knowledge of the spread of disease and the importance of preventive measures in decreasing the risk for COVID-19. Concerns regarding the COVID-19 vaccine should be addressed in the media and awareness campaigns are a must to shed light on the importance of vaccines to prevent COVID-19 infection.

There are a few limitations that should be addressed in the understanding of the results of the current study. First, a major limitation is that the survey was conducted online, which imposes methodological limitations because of the passive exclusion of inactive and non-social media users. Second, this study used a snowball sampling technique that might result in sampling bias; therefore, a study with random sampling is recommended in the future. Though the survey has been performed in the early days of the vaccination rollout campaign, its findings might be different once the mass vaccination program is conducted countrywide targeting the general population as well.

## CONCLUSIONS

The findings of the present study showed an adequate knowledge and positive attitudes of FHWs after getting the COVID-19 vaccination in Pakistan. With this study also providing basic knowledge for confidence building and improvements in communication in relation to COVID-19 vaccines, also highlighted the number of socio-demographic factors influencing the knowledge, attitudes, and practices of FHWs that can give direction to healthcare authorities, policymakers, and public health experts in Pakistan in health services planning. Therefore, this study will be significant in developing COVID-19 vaccination-related awareness and health education programs in the future.

### Funding
The authors received no funding for this work.

### Competing Interests
The authors declare that they have no competing interests.

### Author Contributions
- Sadia Minhas conceived and designed the experiments, performed the experiments, analyzed the data, prepared figures and/or tables, and approved the final draft.
- Aneequa Sajjad conceived and designed the experiments, performed the experiments, analyzed the data, prepared figures and/or tables, and approved the final draft.
- Iram Manzoor conceived and designed the experiments, performed the experiments, authored or reviewed drafts of the article, and approved the final draft.
- Atika Masood conceived and designed the experiments, performed the experiments, prepared figures and/or tables, authored or reviewed drafts of the article, and approved the final draft.

- Agha Suhail conceived and designed the experiments, performed the experiments, authored or reviewed drafts of the article, and approved the final draft.
- Gul Muhammad Shaikh conceived and designed the experiments, performed the experiments, prepared figures and/or tables, and approved the final draft.
- Muhammad Kashif conceived and designed the experiments, performed the experiments, analyzed the data, prepared figures and/or tables, and approved the final draft.

## Human Ethics

The following information was supplied relating to ethical approvals (*i.e.*, approving body and any reference numbers):

Institutional Review Board of Akhtar Saeed Medical and Dental College, Lahore

## Data Availability

The raw data is available in the Supplemental File.

## Supplemental Information

Supplemental information for this article can be found online at http://dx.doi.org/10.7717/peerj.14727#supplemental-information.

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
