# Peer review of "Knowledge, attitude, and practices of front line health workers after receiving a COVID-19 vaccine: a cross-sectional study in Pakistan"

_PeerJ, doi:10.7717/peerj.14727_

## Round 0.1 · original submission · Major Revisions

The manuscript needs major English editing and some points related to the issue raised by the reviewers need to be addressed.

·

Basic reporting

the manuscript and data is valuable.

English language needed proof reading.

References need re-formating.

Experimental design

The manuscript is undere the scope of the journal.

All the section of the manuscript follow basic guidelines.

Validity of the findings

the findings of the study can make an impact on the society.

Conclusion need re-structutring and re-pharisng

Reviewer 2 ·

Basic reporting

The English language needs improvement throughout the manuscript to make it clearly understandable to the international audience. I have indicated a few examples.

The topic of study is of importance during the current pandemic and in possible future pandemics. The introduction is quite lengthy and lacks flow. However, it can be improved. Discussion also lacks flow and seems to drift away from the hypothesis or the objective of the study. I suggest improvement of the tables and associated analysis of the results which will be of help in improving the discussion.

I understand that the figure that has been included in the manuscript is irrelevant to the topic of study. Data presentation in the table can be tremendously improved.
Pls refer to additional comments.

Experimental design

The research falls within the scope of the journal. Even though the research question has been well defined, and methods well described, the presentation of the results and associated discussion needs improvement. The discussion needs to be more focused to relate to the objectives and results of the study.
Pls refer to additional comments.

Validity of the findings

The findings seem valid. However, the tables and the presentation of the results require major improvement. The conclusion also requires modification after reanalysis of the results.
Pls refer to additional comments.

Additional comments

Introduction

Line 60 - World Health Organization should begin with capital letters.
Line 62 and elsewhere - I would suggest using the abbreviation ‘FHW’ for ‘frontline health workers’ throughout the manuscript, rather than repeating them several times.
Line 62 and elsewhere - Worldwide should begin with small letter.
Line 72 - “Ten COVID-19 vaccines have been registered by…….” Pls provide reference.
Line 81 - Pls provide the extended form for ‘sops’, if what you mean is ‘Standard Operating Procedure’. Moreover, SOP should be in capital letters.
Lines 84 and 325 - Pls change ‘Literature research …..’ to ‘Literature search ….’ or ‘Studies have revealed’.
Line 96 - In April 2020 the ‘Access to COVID-19 Tools Accelerator’ was initiated, and not the COVAX. The aim of COVAX is to coordinate equitable distribution of vaccines, test kits, etc.


Methods and Results

Line 198 - The authors have indicated the number of respondents and incomplete forms. Pls provide response rate as well (how many individuals were contacted and how many responded).

The results section is too long. Since all the results are indicated in the tables, you don’t need to comment on each of them. Pls comment on only the important results.

Table 1
1. Other than the listed education levels like MBBS, BDS, etc. what do the authors mean by ‘Graduation’ and ‘Under Graduation’? MBBS or BDS can come under one of these. How are they different?
2. Under “Are you currently in practice”, more than 50% indicated “No”. I understand that this is a source of bias.
3. The question “Where did you receive information about the COVID-19 vaccine?” seems more appropriate in Table 2.

Table 2
1. For the first two questions, pls indicate the percentage of correct responses for each sub-question.
2. Since Table 2 is about frequencies and percentages regarding knowledge about COVID-19 vaccine, questions 3 and 4 (“Have you ever been infected with COVID-19 infection?” and “Do you feel nervous before getting COVID-19 vaccination?”) don’t fit in the table as they do not measure knowledge about COVID-19 vaccines.

Table 3
The last question (“Have you ever had serious reaction after getting COVID-19 vaccine?”) is not an indicator of attitude. Moreover, why did the authors analyze only beliefs and attitude? What about knowledge and practices?

Table 5
Pls explain why you chose to analyze the association of attitudes and beliefs only with gender. What about the other demographics?

Table 6
1. In the regression analyses, why was only attitudes and practices analyzed?
2. I suggest the authors perform bivariate analyses before performing regression analysis.
3. In the column under ‘Variables’, pls explain which variable indicates significant association. For example, in gender, is it male or female? Based on marital status, is it married or single.
4. In the note below the table, what do the authors indicate by the numbering of the variables?

Figure
The figure does not seem relevant to the objective of the study.

Discussion

Line 296 - I suggest changing “This study is going to give first-hand……” to “This study provides first-hand…..”.
Line 302 - Pls correct “Now a day’s”.
Line 333 - “Misconception and lack of trust…..”. The sentence is incomplete. Trust towards what?

Overall, the discussion does not clearly address the objective of the study. Pls modify the discussion after reconsidering the comments on the results.

Conclusion
Line 367 - What do you mean by “livings and lives”? Pls use appropriate language. Pls modify the results after considering the comments on the results.

·

Basic reporting

Some spelling and grammatical errors. Please justify the inclusion of hospital administrative staff as frontline health workers.

Experimental design

The method is commonly used, which is a survey. Questionnaire was used previously in other publication. Some calculations of percentage in table 3 did not add up to 100% and in table 4 more than 100%. Please recheck all the calculations.

Validity of the findings

In general the conclusions are supported by the findings, which are relevant to the specific country Pakistan and possibly worldwide.

Additional comments

I suggest to add more eplanation regarding the impact of these results such as improving policies or others.

---

## Round 0.2 · Major Revisions

This manuscript needs a major revision. Please follow the recommendations proposed by the reviewers when doing the corrections.

Reviewer 2 ·

Basic reporting

The authors claim that professional English language editing has been done. However, the English language is still poor and needs significant improvement.

Experimental design

No major concerns.

Validity of the findings

No major concerns.

Additional comments

General comments:

1. The English language is still poor. It still needs significant improvement.

Introduction:
Lines 103-107: Pls provide reference. Although a reference has been mentioned in the text, it is not available in the reference list.

Materials and methods:
1. Please provide the response rate.

Results:
Table 1: Pls indicate which group of FHWs come under ‘under graduation’.

Table 2: In the rebuttal, the authors argue that “if someone was ever infected with COVID-19 and he is saying Yes, that simply means he had the knowledge of COVID-19 infection.’ Does it mean that those not infected with COVID-19, have no/lower knowledge about it? I understand that knowledge about COVID-19 is not dependent on whether one has been infected with it or not. This is especially true in the case of HCWs. One can be an expert on COVID-19 without prior infection with the virus. Therefore, I understand that the question “Have you ever been infected with COVID-19 infection?” is not a measure of knowledge.

Similarly, in table 2, the authors argue that “someone who feels nervous before COVID19 vaccine can have lower knowledge related to it”. Does it mean that those who are not nervous about COVID-19 vaccines have better knowledge about it? It can also be vice versa. Pls consider my previous comment regarding my concern about this question being used as an indication of knowledge about COVID-19 vaccines.

Table 6: In the column under ‘Variables’, pls explain which variable indicates significant association. For example, in gender, is it male or female? Based on marital status, is it married or single. The authors have replied that p values have already been indicated along each variable. However, it is not clear which sub-variable, for example, which age group or which gender, shows significance.

Table 6: In the note below the table, what do the authors indicate by the numbering of the variables? The authors have replied “As the nominal variables were entered in regression, their response categories/ attributes are mentioned so the results can make sense to the reader”. Pls indicate as to how the reader can make sense as to which age group or which gender shows significance?

Discussion:
1. Lines 296-297: The information provided needs to be rechecked. In all three studies mentioned in the referred article, more than 85% (not 11%, as indicated by the authors) of the Australian population was willing to get vaccinated against COVID-19.
2. Line 348: ‘Limitations’ should begin as a separate paragraph.
3. Line 351: What do the authors mean by “though extensive use of social media will reveal such concerns”.
4. Line 352: What do the authors mean by “might result in the selection of biasness”.

Conclusions:
Not focused. Needs improvement.

·

Basic reporting

Some minor editing is required such as spacing between words - lines 30, 33, and throughout the text. Line 615 - the word "was" was repeated twice

Experimental design

Please explain this statement "The biasness of the study was decreased by keeping the survey open for 3 months." Is there any reference(s) when prevalence was estimated at 50%?

Validity of the findings

I feel that the statement in the abstract "while practices were significantly predicted from marital status, the number of information sources, knowledge of COVID-19 vaccination, and attitude after getting COVID-19 vaccination." is confusing. Please rewrite if possible.

Additional comments

I personally feel that the title of this paper should be "The effects of covid-19 vaccination on FHW in Pakistan regarding knowledge, attitude, and practices". I am supporting the publication of this paper with the view that FHWs are important in influencing the general public in accepting any health related measures such as vaccines, oral hygiene care and others.

---

## Round 0.3 · Minor Revisions

Please address reviewer’s 1 comment related to response rate of the questionnaires.

Reviewer 2 ·

Basic reporting

Acceptable.

Experimental design

Acceptable.

Validity of the findings

Valid.

Additional comments

In the rebuttal, the authors mentioned that the the response rate has already been mentioned in the 'Results' section, lines 190-191. What has been mentioned there is just the percentage of respondents included in the study after exclusion of incomplete survey questionnaires. I would suggest the authors to look up as to what I mean by 'response rate', and provide the percentage.

·

Basic reporting

No comment

Experimental design

No comment

Validity of the findings

No comment

Additional comments

I am ok to recommend acceptance and publication of this manuscript

---

## Round 0.4 · accepted · Accept

The corrections have been made. So the manuscript is now accepted for publication.

Reviewer 2 ·

Basic reporting

OK

Experimental design

OK

Validity of the findings

OK

Additional comments

No Comments